# Identifying the Risk Regions of Wild Boar (*Sus scrofa*) Incidents in China

**DOI:** 10.3390/ani13203186

**Published:** 2023-10-12

**Authors:** Boming Zheng, Xijie Lin, Xinhua Qi

**Affiliations:** 1Institute of Geography, Fujian Normal University, Fuzhou 350007, China; qbx20210132@yjs.fjnu.edu.cn (B.Z.); qsx20211174@yjs.fjnu.edu.cn (X.L.); 2School of Geographical Sciences, Fujian Normal University, Fuzhou 350007, China; 3Key Laboratory for Humid Subtropical Eco-Geographical Processes of the Ministry, Fujian Normal University, Fuzhou 350007, China

**Keywords:** human–wild-boar conflict, risk region, identify, Maxent, China

## Abstract

**Simple Summary:**

Identifying the risk regions of human–wildlife conflict is a pertinent topic in current human–wildlife conflict research. Most of the existing studies obtained their wildlife incident information through field surveys to identify risk regions. However, field surveys require a lot of personnel, time, and money. Therefore, they are generally used for meso- and microscale studies as they are difficult to apply to macroscale explorations. In this paper, we attempted to expand the data sources of wildlife incident information to enable macroscale wildlife incident risk region identification more efficiently and at a lower cost. We chose wild boar incidents in China as an example and used web crawling technology to obtain reports of wild boar incidents from internet databases. We then extracted the spatial location information of wild boar incidents from the reports. Subsequently, a system of environmental variables was constructed. Finally, the Maxent model, which provides predictions with higher accuracy and requires less location information, was applied to identify the risk regions of wild boar incidents in China. We observed that approximately 12.18% of China was at a high-risk level, mainly on the eastern side of the Huhuanyong Line. The risk of wild boar incidents was related to the climate, landscape, and topography as well as human disturbance. Variables such as the annual precipitation, GDP index, mean annual temperature, distance from forestland, distance from cultivated land, and elevation strongly influenced the risk of wild boar incidents.

**Abstract:**

The objectives of this study were to identify the risk regions of wild boar incidents in China and to draw a risk map. Risk maps can be used to plan the prioritization of preventive measures, increasing management effectiveness from both a short- and a long-term perspective. We used a web crawler (web information access technology) to obtain reports of wild boar incidents from China’s largest search engine (Baidu) and obtained 196 valid geographic locations of wild boar incidents from the reports. Subsequently, a system of environmental variables—with climate, topography, landscape, and human disturbance as the main variable types—was constructed, based on human–land-system thinking. Finally, the Maxent model was applied to predict the risk space of wild boar incidents in China by integrating the geographic location information for wild boar incidents with the environmental variables. We observed that the types of environmental variables that contributed to wild boar incidents were in the descending order of climate (40.5%) > human disturbance (25.2%) > landscape (24.4%) > topography (9.8%). Among the 14 environmental variables, annual precipitation, the GDP index, and the mean annual temperature were the main environmental variables. The distance from woodland, distance from cultivated land, and elevation were the secondary environmental variables. The response curves of the environmental variables demonstrated that the highest probability of wild boar incidents occurred when the annual average temperature was 16 °C, the annual precipitation was 800 mm, and the altitudes were 150 m and 1800 m. The probability of wild boar incidents decreased with an increase in the distance from cultivated and forested land, and increased sharply and then levelled off with an increase in the GDP index. Approximately 12.18% of China was identified as being at a high risk of wild boar incidents, mainly on the eastern side of the Huhuanyong Line.

## 1. Introduction

Economic development and population growth inevitably increase human demand for land resources, severely encroaching on wildlife habitats [1,2]. This leads to an increasing overlap between human activity space and wildlife habitats, triggering frequent incidents between humans and wildlife [3]. Incidents between humans and wildlife have serious negative impacts on human production and daily life and also weaken the interest local residents have in biodiversity conservation [4]. In certain cases, this can trigger the retaliatory killing of wildlife [5,6]. How to alleviate and eliminate human-wildlife conflicts (HWCs), reconcile human livelihood welfare and wildlife survival rights, and build a human-wildlife living community have become important issues to be solved in global biodiversity conservation work.

Incidents between humans and wildlife are essentially a spatial and temporal overlap of human and wildlife activities [7]. These may be negative human-wildlife interactions (HWIs) that often have many tangible and intangible impacts on humans, including personal injury and death, crop damage, and livestock depredation as well as negative effects on food security and mental health [8,9,10,11,12]. Research on incidents between humans and wildlife has intensified under the increasingly serious phenomenon of incidents between them. Theoretical research perspectives have been expanded; theories and methods from different disciplines such as sociology [13], economics [14], geography [15], psychology [16], and ecology [17] have been widely applied; and fruitful results have accrued. Through the cross-fertilization of disciplines, academics have deepened their understanding of spatial distribution patterns [18], accidental damage [19], mitigation measures [20], conflict assessment [21], and compensation mechanisms [22].

The spatial distribution patterns of incidents between humans and wildlife—ascertained based on geographic methods and techniques—have been a pertinent research topic in recent years. The spatial scales of these studies can be broadly divided into two levels: microscale and macroscale. By exploring microenvironmental characteristics and macroscopic hotspot clustering in the spatial distribution of incidents between humans and wildlife, scholars can provide scientific references for the management and prevention of incidents between humans and wildlife. Microscale studies focus on exploring the environmental characteristics of spatial distribution. Related research has demonstrated that the probability of human–bear (*Ursidae*) conflict is negatively correlated with the distance of the conflict site from the core of the reserve [23], the probability of livestock depredation by predators is higher in areas close to villages and open habitats [24], and the probability of human–wild-boar (*Sus scrofa*) conflict is positively correlated with the distance to the nearest water source [25]. HWC is often closely related to the distance from human communities (farmland, tea plantations, livestock farms, etc.), wildlife habitats (forests, water sources, etc.), and biodiversity-rich areas (protected areas, national parks, etc.). Macroscopic spatial studies mainly investigate the hotspots of regional HWC events, commonly in conjunction with large-scale periods. For example, Baruch-Mordo used the Getis-ord Gi* spatial clustering statistic to describe the location and assess the magnitude of human–bear conflicts in Colorado from 1986 to 2003 [26]. Basak analyzed 2512 incidents in which animals (of which 85% were mammals and 15% birds) were involved in conflict situations between 2007 and 2013 and identified spatial clusters of conflict [27].

The existing spatial studies on incidents between humans and wildlife mostly distill the laws and summarize the current situation; there are relatively few studies on the prediction of future potential risk spaces. Most prediction studies obtain spatial information on incidents between humans and wildlife through field surveys and develop predictions based on this information. However, field surveys are often labor-intensive, time-consuming, and costly. This results in the risk of spatial prediction studies being limited to a microscale; thus, it is difficult to obtain macropredictions. Methods to expand access to spatial information on incidents between humans and wildlife and to realize the spatial prediction of the macroscopic risk of incidents between humans and wildlife are issues that require further exploration.

In this study, we selected Chinese wild boar incidents as an example. We obtained historical spatial information of wild boar incidents in China using web crawler technology, selected the environmental factors, and finally applied the maximum entropy model (Maxent) to predict the potential risk space of wild boar incidents in China. China is the third largest country in the world in terms of land area, with a rich biodiversity and large population. Approximately one-third of the land is at a high risk of conflict between biodiversity conservation efforts and human activities. Among the many types of incidents between humans and wildlife in China, wild boar incidents frequently occur. In China’s Sichuan Province, as many as 7000–8000 wild boar damage incidents are reported every year and the economic losses caused by wild boar crop raiding are approximately USD 33 million [28]. Wild boars have a strong reproductive ability and, under a policy of biodiversity conservation in China, their population number and scale are continuously expanding. With their strong adaptability for survival, they are widely distributed in various regions of China. When their complicated diet and ferocious nature are coupled, they often destroy crops and hurt humans. This has a serious negative impact on human production and life. To mitigate the hazards of wild boars, the China Forestry and Grassland Bureau performed comprehensive pilot projects in 2017 to prevent and control the hazards of wild boars in 14 provinces and regions; each pilot area has successively introduced working programs and policies on prevention, hunting, compensation, and insurance. In 2021, the China Forestry and Grassland Administration issued the Circular of the State Forestry and Grassland Administration on Further Improving the Prevention and Control of Wild Boar Hazards and the Technical Points for Prevention and Control of Wild Boar Hazards, guiding localities to strengthen their prevention and control of wild boar hazards. We hope that our study provides a scientific reference for the differentiated management and proactive prevention of wild boar incidents in China, helping to expand and enrich the scale and perspective of the spatial studies of incidents between humans and wildlife.

## 2. Data and Methods

### 2.1. Data

#### 2.1.1. Wild Boar Incident Locations

Internet technology is currently highly developed and popularized, forming a stable and rich information base. Web crawling technology is regularly updated, enabling required information to be accurately and quickly located and acquired. In this study, we obtained the spatial location information of wild boar incidents in China based on the mode of “keyword (index) + search engine (information base) + web crawling (technology)”. This may provide a reference for the data acquisition of related studies in the future. We used the Baidu search engine (the largest and most commonly used search engine in China; https://www.baidu.com, accessed on 8 May 2022) with keywords such as “wild boar incidents”, “wild boar damage”, and “wild boar injuries”. We used Houyi collector 4.0.1 (web crawler software; https://www.houyicaiji.com, accessed on 8 May 2022) to collect nationwide reports on wild boar incidents. Collating the reports revealed a timeframe covering the period from 2008 to 2022. The specific spatial location information for the wild boar incidents was extracted from valid report contents. The corresponding decimal formats of latitude and longitude were obtained using Baidu Maps (an intelligent location service platform; https://map.baidu.com, accessed on 8 May 2022). To avoid overfitting, the SDM Tools function in ArcGIS 10.8 [29] was applied to check the spatial autocorrelation of the extracted geographic locations, ensuring that each raster had only one distribution location (the raster size was 1 km) and that the distance between every two locations was greater than 10 km. Finally, 196 valid locations were retained (Figure 1). The latitude and longitude information was organized into an Excel sheet and then imported into ArcGIS and saved as CSV format files (supported with the Maxent model) for the subsequent analysis. The 196 locations were mainly in the central and eastern regions, with a greater distribution in the Gansu, Shaanxi, Hubei, and Sichuan provinces. Only 17 of these 196 locations involved wild boar attacks on humans. The remaining 179 locations were property damage incidents, the vast majority of which involved wild boars trampling farmland and damaging crops. Wild boar attacks on people are the result of encounters between people and wild boars; as the majority of property damage incidents occurred in locations with high human activity (e.g., farmland, tea plantations, etc.), there was also an increased risk of wild boar attacks in these locations. Therefore, we uniformly defined 196 locations as wild boar incident point inputs in the model.

#### 2.1.2. Environmental Variable Data

In this study, a total of 14 environmental variables were selected in four categories: climate, topography, landscape, and human disturbance (Table 1). Climate, topography, and landscape reflected the natural factors and human disturbance reflected the human activities. Annual average temperature and annual precipitation data were gathered from the World Climate Database (http://www.worldclimate.com, accessed on 24 June 2022). The altitude data were obtained from the scientific database of the Chinese Academy of Sciences (https://www.cas.cn/ky/kycc/kxsjk, accessed on 24 June 2022). The slope data were calculated based on digital elevation model (DEM) data by applying ArcGIS. The data for land-use types (such as cultivated land, forestland, water source, and grassland) were obtained from the Geographic Monitoring Cloud Platform (http://www.dsac.cn, accessed on 24 June 2022) and the distance of each image element from the nearest land-use type was measured using the Euclidean distance analysis method in ArcGIS. Normalized difference vegetation index (NDVI) data were obtained from the Geospatial Data Cloud website (http://www.gscloud.cn, accessed on 24 June 2022). The county boundaries and major road data were obtained from the National Basic Geographic Information Center (http://www.ngcc.cn, accessed on 25 June 2022) and the distance of each image element from the nearest county boundary and road was also measured with the Euclidean distance analysis method in ArcGIS. GDP data were obtained from the Data Center for Resource and Environmental Sciences, Chinese Academy of Sciences (http://www.resdc.cn, accessed on 24 June 2022). Population density data were obtained from the Worldpop website (https://www.worldpop.org, accessed on 24 June 2022). The spatial resolution of all environmental data was 1 km.

To avoid problems such as autocorrelation and multicollinearity among the environmental variables that could lead to high AUC values of the model output [8], the Band Collection Statistics tool in ArcGIS was applied to check the correlation of the environmental variables. If the absolute value of the correlation between two variables was greater than 0.85 [30], only one factor was selected to be entered into the model. The final results revealed that the absolute values of the correlation coefficients between all variables were less than 0.85; thus, all variables were included in the model (Table 2). Finally, ArcGIS software was applied to unify the raster size, the coordinate system was set to GCS_WGS_1984, and the output was in an ASC format supported with the Maxent model.

### 2.2. Methods

#### 2.2.1. Maxent Model

The maximum entropy model (Maxent) is a well-established model for the prediction of the potential range of a species in a target area by generalizing or simulating the possible distribution of the maximum entropy of the species using mathematical models based on a known limited number of species distribution points and the particular survival environment of the species [31]. Compared with species distribution models [32], generalized regression analyses and spatial predictions [33], generalized additive models [34], and the genetic algorithm for rule-set production [35], the Maxent model requires less data and can produce predictions with higher accuracy and fewer known species distribution locations [36]. Since Phillips wrote and developed the model using JAVA language in 2004 [37], it has been widely used in many studies, including species habitat predictions [38], invasive alien species [39], climate change effects on species distributions [40], forest pests [41], and fire-risk predictions [42]. In this study, the Maxent model was applied to the field of wildlife incident studies to obtain the macroscale risk of the spatial predictions of incidents between humans and wildlife.

#### 2.2.2. Data Analysis

The location of the spatial distribution of wild boar incidents and the environmental variables were inputted into Maxent 3.4.4 software for the calculations [43]. From the 196 samples, 75% were randomly selected as the training set for the model construction and the remaining 25% were used as the test set for the model testing. Bootstrap was selected as the repetition run category. The number of model iterations and repetitions was set to 10,000 [44] and 10, respectively, and the remaining parameters were set using the software default [44]. The area under the receiver operating characteristic curve (AUC) was used to evaluate the model accuracy. AUC values range from 0 to 1. The larger the value, the higher the prediction accuracy. Generally, a value below 0.5 indicates unreliable prediction results, between 0.7 and 0.8 is moderate, between 0.8 and 0.9 indicates reliable results, and above 0.9 is considered to be excellent [45]. The jackknife method was used to analyze the importance of each environmental variable. The simulation results were saved in an ASC format by selecting the Cloglog output method [46]. The output results of the Maxent model were imported into ArcGIS and converted into raster files to obtain a spatial distribution map of wild boar incident risks in China. The natural breaks classification (Jenks) method in ArcGIS was applied to classify the risk space of wild boar incidents in China into 5 categories. These were high (0.500–0.908), sub-high (0.335–0.499), medium (0.193–0.335), sub-low (0.065–0.192), and low (0–0.064) risk spaces. The distribution of different risk level spaces was analyzed and their areas were calculated.

## 3. Results

As shown in Figure 2, the average training set AUC value of the Chinese wild boar incident risk space identification model was 0.957. This placed it in the excellent category, indicating that the Chinese wild boar incident risk space that was predicted based on the Maxent model had high accuracy and confidence.

### 3.1. Environmental Variables

#### 3.1.1. Contribution of Environmental Variables

The contribution of the 14 environmental variables to the prediction of wild boar incidents was determined based on the Maxent model, and the combined contribution of the four major variables of climate, topography, landscape, and human disturbance was analyzed (Table 3). Overall, the climate variable contributed the most (40.5%), followed by human disturbance (25.2%), landscape (24.4%), and topography (9.8%). Annual precipitation (X2), the GDP index (X13), and annual average temperature (X1) were the top three environmental variables in terms of contribution (26.6%, 23.7%, and 13.9%, respectively), with a cumulative contribution of 64.2%. The contributions of distance from forestland (X6), distance from cultivated land (X5), and altitude (X3) were 7.7%, 7%, and 5.5%, respectively. The cumulative contribution of the above six environmental variables to wild boar incidents reached 84.4%. The contribution of the remaining eight environmental variables did not exceed 5%, indicating that the effect of these factors on wild boar incidents was not significant. Figure 3 characterizes the main AUC values when each environmental factor was individually used to construct the model. The results revealed that the order of the variables for the top six AUC values was X13 > X2 > X1 > X14 > X5 > X10. Four of the six environmental variables with significant contributions (X2, X13, X1, and X5) also ranked in the top six in terms of AUC values. Two other variables, X6 and X3, also ranked at positions 7 and 8, respectively, further validating the importance of these factors.

#### 3.1.2. Dynamic Response of Environmental Variables

The six main environmental variables of annual average temperature, annual precipitation, altitude, distance from cultivated land, distance from forestland, and GDP index demonstrated four different types of response curves (Figure 4). The response curves of the annual average temperature and annual precipitation first increased and then decreased (Figure 4a,b). The probability of wild boar incidents was higher in areas where the annual average temperature was in the range of 11–20.5 °C, with the highest probability at 16 °C (Figure 4a). The probability of wild boar incidents was higher in areas with annual precipitation in the range of 600–1600 mm, with the highest probability at 800 mm (Figure 4b). The altitude response curve had bimodal characteristics, with a higher probability of wild boar incidents in areas where the altitude was in the 0–450 m and 1700–2000 m ranges; the highest probability of wild boar incidents was in areas where the altitude was 150 m and 1800 m (Figure 4c). The response curves of the distance from cultivated land and forestland revealed a decreasing trend (Figure 4d,e). The further the distance from cultivated land and forestland, the lower the probability of wild boar incidents. The GDP index response curve demonstrated that the probability of wild boar incidents sharply increased and then plateaued with an increase in the GDP index (Figure 4f).

### 3.2. Distribution of the Risk Space of Wild Boar Incidents

The spatial locations of wild boar incidents and the environmental variables were integrated to generate a spatial distribution map of wild boar incident risks in China based on the Maxent model. The natural breaks classification (Jenks) method was applied to classify the risk spaces into five levels (Figure 5). Statistically, the areas of each risk level of wild boar incidents in the country were high risk (393,524 km^2^), sub-high risk (780,566 km^2^), medium risk (979,365 km^2^), sub-low risk (780,566 km^2^), and low risk (6,343,719 km^2^). Approximately 12.18% of the area was in the sub-high and high levels of risk. The Huhuanyong Line is the dividing line of population density and economic development levels in China; the population density and economic development levels on its eastern side are much higher than those on the western side [47]. Overall, the risk space of wild boar incidents in China revealed a significant polarization with the Huhuanyong Line as the boundary. The sub-high- and high-risk areas were mainly distributed on the east side of the Huhuanyong Line, whereas the west side was dominated with low-risk areas. From a regional perspective, the areas with sub-high and high risks of wild boar incidents were mainly distributed in the Qinling Mountains, Wuyi Mountains, Changbai Mountains, North China Plain, and middle and lower reaches of the Yangtze River Plain. Specifically, the top ten provinces in terms of areas of sub-high- and high-risk spaces were Shaanxi, Sichuan, Hubei, Jiangxi, Shanxi, Hunan, Guizhou, Yunnan, Henan, and Shandong. The top ten provinces in terms of areas of sub-high- and high-risk spaces as a percentage of the provincial area were Shaanxi, Zhejiang, Shanxi, Jiangxi, Hubei, Chongqing, Guizhou, Shandong, Anhui, and Henan (Appendix A).

## 4. Discussion

Our research expanded the data sources of spatial information on incidents between humans and wildlife. We constructed a system of environmental variables that considered human activities and natural landscapes. The results of the model accuracy test were in the excellent category, indicating that the spatial prediction model we constructed for the risk of wild boar incidents in China had high credibility and accuracy. The identification of risk spaces can help with the differentiated management of incidents between humans and wildlife.

### 4.1. Expansion of Data Sources

Existing studies based on the Maxent model mostly focus on the prediction of potential and suitable habitats for plants and animals [38]. The spatial location information of samples is often obtained through field surveys [48], existing databases [49], and geographic information technology [50]. Several studies have used the model to predict the risk spaces for incidents between humans and wildlife [51,52]. This modeling requires information on specific locations of human–wildlife conflicts; most of the existing studies used field survey methods to obtain relevant data [8]. However, field surveys often impose high demands on personnel and funding. This leads to existing study areas being limited to mesoscale or microscale approaches. Discovering a method to explore the spatial point information of macroscopic-scale incidents between humans and wildlife at a low cost and with high efficiency is an important issue that requires further study. Our research provides potential solutions to this problem, i.e., to obtain wildlife incident information by combining web-based information repositories and web-based information crawling techniques. Predictions with higher accuracy might be possible if wild boar densities could be fitted as weights [53,54]. We were not able to realize this concept because systematic monitoring data for wild boars are not yet available in China. This is an area for future improvements. We have always believed that the most fundamental method to obtain the most complete information on wildlife incidents is to build a systematic statistical monitoring system of wildlife incident information at national and regional levels.

### 4.2. Environmental Variable System Construction

Incidents between humans and wildlife are the result of an overlap between wildlife living space and human production and living space [3]. When selecting environmental variables, a human–land-system approach should be used to account for natural factors and human activities. Jiang considered the influence of topography, landscape, and human disturbance on wild boar incidents when predicting the risk space of wild boar incidents in Jiangxi, but ignored the influence of climate [55]. Several studies have argued that the impact of climate on human–wildlife conflicts should not be underestimated [56]. We added a climate variable to the environmental variable system and constructed a four-dimensional environmental variable system of “climate–topography–landscape–human-disturbance”. Climate, topography, and landscape were used to characterize natural factors and human disturbance was used to characterize human activities. Our results revealed that the contribution of climate ranked first among the four environmental types, indicating that climate had a significant influence on the wild boar incidents and should not be ignored. Two climatic variables, annual precipitation and annual average temperature, contributed 26.6% and 13.9%, respectively, to wild boar incidents. The physical comfort needs of wild boars cause them to move to areas with suitable climates. Crops and plants are important food sources for wild boars and climate conditions directly affect the growth and distribution of crops and plants; this may indirectly affect the range and frequency of the activities of wild boars [57,58]. The factor with the highest contribution from the topographic environment category was altitude; the probability of wild boar incidents was higher in areas with an altitude in the range of 0–450 m and 1700–2000 m, indicating that the prevention of wild boar incidents in middle- and low-altitude areas should be strengthened in the future. The factors that had a greater contribution to the landscape environment variable were the distance from cultivated land and distance from forestland; the results revealed that the closer the distance from cultivated land and forestland, the higher the probability of wild boar incidents. This was similar to the conclusion of Jiang [55] and may be related to the living environment and foraging behavior of wild boars. Wild boars commonly live in forests; thus, when a place is closer to a forest, there is a greater likelihood of encountering wild boars [59]. Wild boars prefer to feed on crops, making crop damage the most common wild boar incident [60]. Among the human disturbance types, the GDP index contributed the most. The probability of wild boar incidents was higher at locations with higher GDP indexes. The results of the risk spatial identification also revealed a higher risk of wild boar incidents in the area east of the Huhuanyong Line. These results suggest that the risk of human–wild-boar conflict may be higher when the regional population density and levels of economic development are higher. Human demand for various resources increases with human economic development. The demand for resources drives human activity space to continuously squeeze and fragment wildlife habitats, resulting in a high degree of overlap between human activity space and wildlife habitats. This eventually leads to frequent human-wildlife conflicts [61]. This does not mean that the western region, with its lower population density and levels of economic development, was free from the risk of wild boar incidents—only that the risk was relatively low.

The construction of a system of environmental variables for incidents between humans and wildlife also depends on differences in the species and regions of concern. For example, wild boars mainly damage crops; therefore, the distance to cultivated land should be included in the environmental variables. Carnivores such as wolves (*Canis lupus*), snow leopards (*Panthera uncia*), and bears prey on livestock; therefore, environmental variables related to livestock should be considered [62,63].

### 4.3. Management Recommendation

Risk maps can be used to plan the prioritization of preventive measures to increase management effectiveness from both short- and long-term perspectives [25]. According to our risk map, relevant preventive measures could be prioritized for trial implementations in Shaanxi, Sichuan, Hubei, Jiangxi, Shanxi, Hunan, Guizhou, Yunnan, Henan, and Shandong on the eastern side of the Huhuanyong Line and extended to other lower-risk areas after proof of their effectiveness. This would be less costly and more efficient. In higher-risk areas, we suggest that the following measures are used to prevent wild boar incidents. First, the systematic monitoring of wild boar populations and a statistical system of wild boar incidents should be implemented. This will increase knowledge of the status of wild boar population developments to ensure that hunting operations are implemented in a planned manner to control the population [25]. It will also enrich and improve the database of the wild boar risk space identification model, identifying risk spaces with higher accuracy [53,54]. Second, as wild boar incidents are more likely to occur in areas close to forests, cultivated land, and densely populated areas, preventive measures can be selectively applied in these locations. Tying colorful ribbons or sarees [64], using white-colored flying/flashing ribbons [65], and hanging glass bottles and stones freely and closely (so that they collide from wind motion) provide wild boars with the false indication of a human presence. This may minimize the frequency of their visits. Using noise/fire deterrents or trained dogs to scare away animals [65,66], installing early warning systems and electric fencing, spraying a feral pig dung solution, smoking dried dung cakes, spreading pieces of human hair in croplands [64], planting xerophytic plants or thorny bushes around cultivated areas [66,67], and planting crops that wild boars do not like to eat are other non-invasive methods to tackle HWCs. Finally, we recommend the implementation of compensation policies and insurance schemes to reduce the economic losses caused by wild boars; the criteria for compensation should be tailored to local conditions.

### 4.4. Suggestions for Future Research

We aim to continue to expand the data sources in the study of the spatial identification of the risk of incidents between humans and wildlife. Through the integration of multiple data sources, we may achieve predictions with higher accuracy to enhance the research on the spatial distribution of wildlife causing damage in relevant areas. Only wild boars were used as the research object in this study; future research could expand on the number of species involved in studied incidents. If all incident-involved species can be used as research objects to achieve risk space identification, it will have significance for the prevention of and compensation for incidents between humans and wildlife.

## Figures and Tables

**Figure 1 animals-13-03186-f001:**
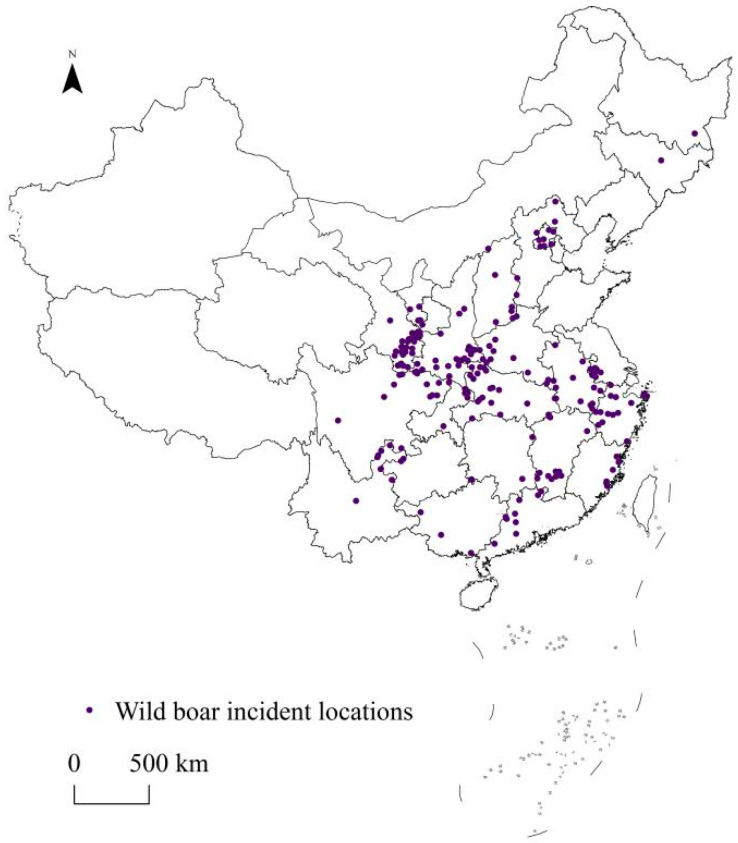
Spatial visualization of the collected wild boar incident locations.

**Figure 2 animals-13-03186-f002:**
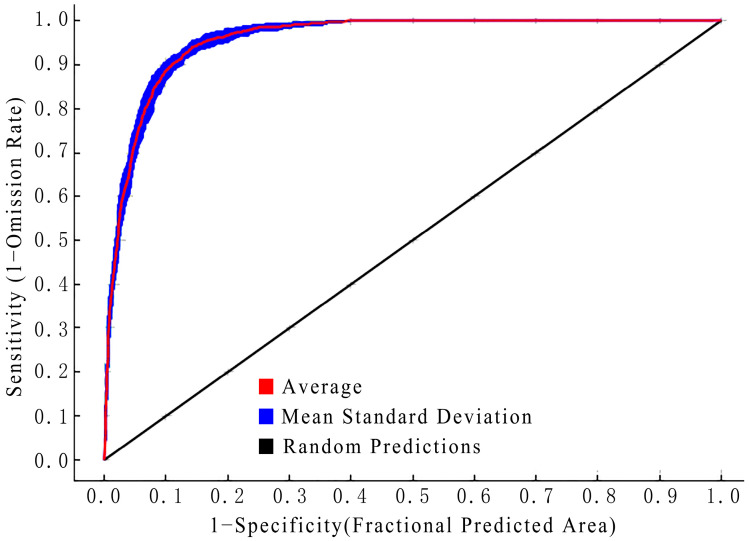
Accuracy test of the Maxent model.

**Figure 3 animals-13-03186-f003:**
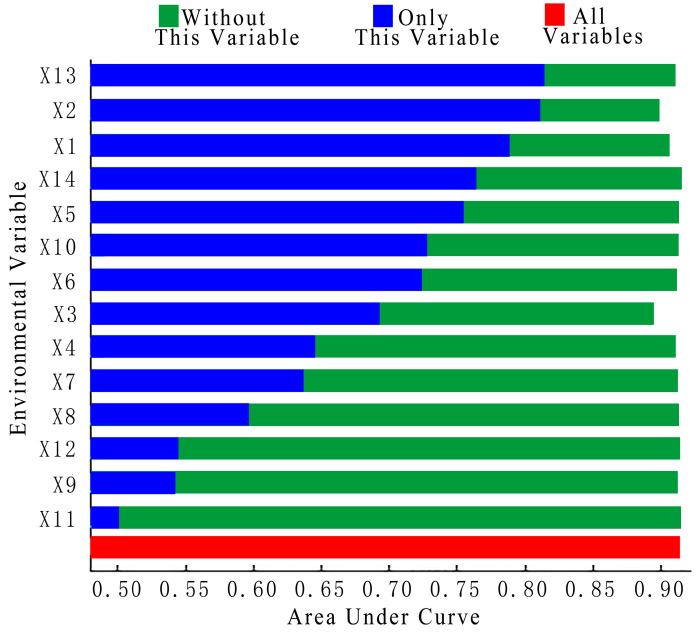
Jackknife testing of importance of variables.

**Figure 4 animals-13-03186-f004:**
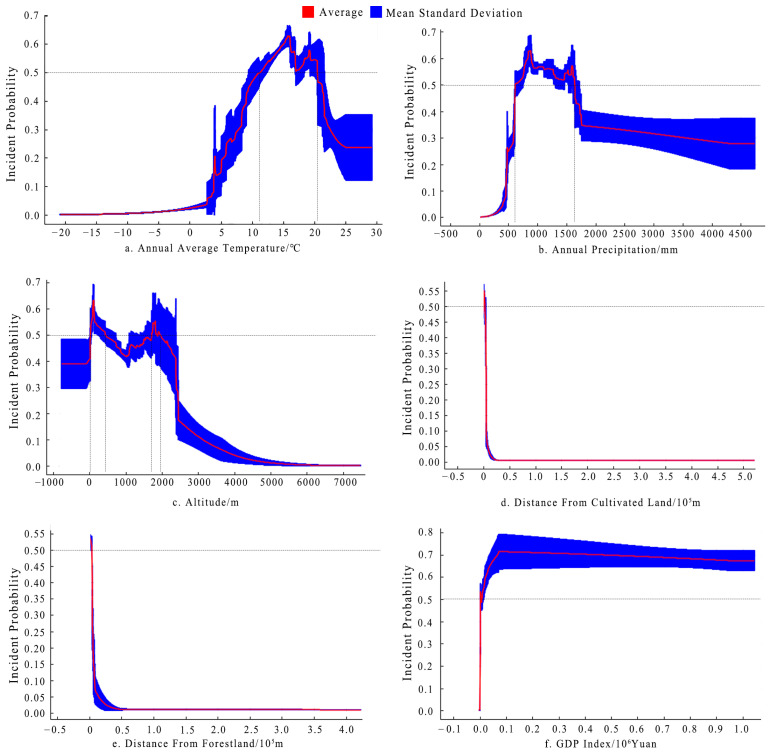
Response curves of the main environmental variables with large contribution rates to the probability of wild boar incidents. The results of the natural breakpoint method state that the probability of a wild boar incident is high if it is above 0.5; we also used 0.5 as the threshold value.

**Figure 5 animals-13-03186-f005:**
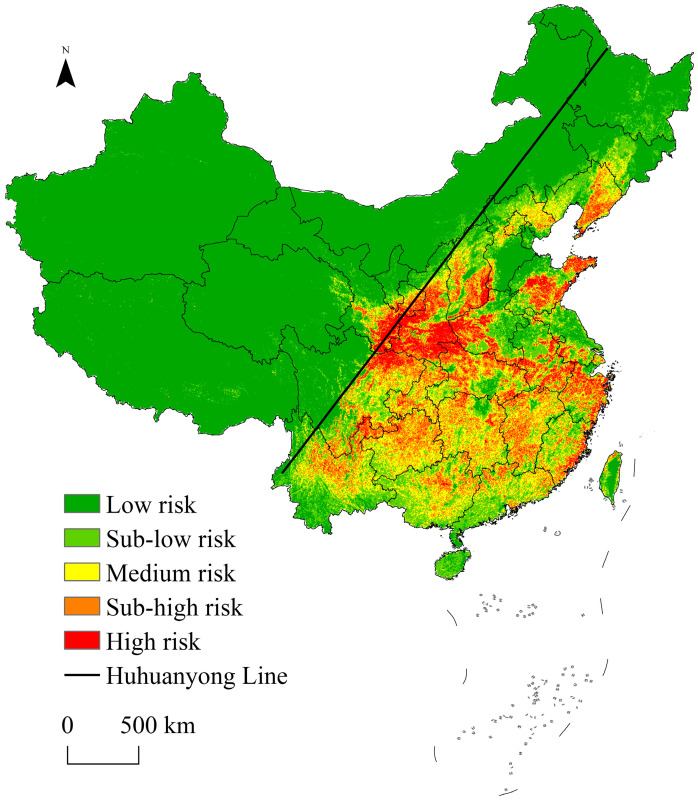
Risk space identification of wild boar incidents in China.

**Table 1 animals-13-03186-t001:** Selection of environmental variables.

Type	Variable	Number
Climate	Annual average temperature	X1
	Annual precipitation	X2
Topography	Altitude	X3
	Slope	X4
Landscape	Distance from cultivated land	X5
	Distance from forestland	X6
	Distance from water source	X7
	Distance from grassland	X8
	Vegetation type	X9
	NDVI	X10
Human disturbance	Distance from county boundary	X11
	Distance from road	X12
	GDP index	X13
	Population density	X14

**Table 2 animals-13-03186-t002:** Correlation between environmental variables.

	X1	X2	X3	X4	X5	X6	X7	X8	X9	X10	X11	X12	X13	X14
**X1**	1.00	0.64	−0.71	−0.14	−0.49	−0.44	0.01	0.26	−0.03	0.35	0.02	−0.14	0.13	0.17
**X2**		1.00	−0.38	0.17	−0.38	−0.37	−0.28	−0.01	−0.10	0.68	0.03	−0.25	0.12	0.14
**X3**			1.00	0.38	0.61	0.55	−0.06	−0.28	0.00	−0.43	−0.01	0.17	−0.10	−0.13
**X4**				1.00	−0.04	−0.04	−0.09	−0.20	−0.14	0.13	0.00	−0.09	−0.05	−0.07
**X5**					1.00	0.71	0.12	−0.02	0.05	−0.51	−0.01	0.36	−0.06	−0.08
**X6**						1.00	0.09	0.03	0.07	−0.50	0.00	0.29	−0.05	−0.06
**X7**							1.00	0.28	0.06	−0.32	0.00	0.27	−0.06	−0.08
**X8**								1.00	0.02	−0.08	0.01	0.18	0.05	0.08
**X9**									1.00	−0.12	0.00	0.04	−0.01	0.01
**X10**										1.00	0.00	−0.36	0.03	0.05
**X11**											1.00	0.00	0.02	0.01
**X12**												1.00	−0.03	−0.05
**X13**													1.00	0.44
**X14**														1.00

Bold indicates the environmental variables listed in Table 1.

**Table 3 animals-13-03186-t003:** Contribution of environmental variables and their ranking.

Variable	Variable Contribution (Rank)	Type
Annual precipitation (X2)	26.60% (1)	Climate
GDP index (X13)	23.70% (2)	Human interference
Annual average temperature (X1)	13.90% (3)	Climate
Distance from forestland (X6)	7.70% (4)	Landscape
Distance from cultivated land (X5)	7.00% (5)	Landscape
Altitude (X3)	5.50% (6)	Topography
Slope (X4)	4.30% (7)	Topography
Vegetation type (X9)	4.30% (7)	Landscape
Distance from water source (X7)	2.40% (8)	Landscape
Distance from grassland (X8)	1.90% (9)	Landscape
Population density (X14)	1.50% (10)	Human interference
NDVI (X10)	1.10% (11)	Landscape
Distance from county boundary (X11)	0.00% (12)	Human interference
Distance from road (X12)	0.00% (12)	Human interference

## Data Availability

The data presented in this study are available upon request from the corresponding author.

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
