# Peer review of "Identifying the Risk Regions of Wild Boar (Sus scrofa) Incidents in China"

_animals, 2023, doi:10.3390/ani13203186_

Round 1
Reviewer 1 Report
Overall evaluation:
1. The Authors used a cost-efficient and state-of-the art method to find and map the risk regions of wild boar related damages and incidents throughout China. The studied issue is a worldwide problem; therefore any new and effective ways that help to mitigate human-wildlife conflicts is worth to study and to publish in this Journal. I find the introduction and the method itself very catchy. Similar to other machine learning models, some results can be hard to interpret without decent domain knowledge and this is clearly visible in the discussion chapter, which is the weakest point of the paper from my opinion. Authors were able to synthetize some results from this black box model but I really miss the key message that give practical and useful solutions for the future based on the actual results.
2. I also missed some data about the wild boar density, and wonder why the model was fitted without considering the actual wild boar density and human population as weights when the model was constructed.
3. Provide some information about the distribution of these human vs wild boar incident reports: How many of them were direct attacks or physical conflict with humans and how many resulted in damages to property?
Please see my detailed comments below that hopefully increase the level of the manuscript.
Specific comments about the figures and tables
Figure 3: The title of the y-axis contains a typo: “Environmental Variale” should be Environmental Variable. Visualize the factors in a descending order by their value! (x13-x2-x1 etc). More explanation needed for this figure. What are we seeing, what indicates the importance of the factor etc. The term should be called as "Jackknife testing of variable importance..."
Figure 4: The negative side of the x axis has no meaning in the case of precipitation, distance from cultivated land/forest and GDP. These should be truncated to better visualize the descending shape of the response curves. Figures should be indicated with letters that help referring to them in the text. What was the metric and the threshold that separated "large" from "small" contribution? (mention this in the caption).
Table 4: I find this table unnecessary here, it would be better to place it in the supplementary materials, or only report the percentages (subhigh and high) with human population density and introduce Regions in a descending order by risk.
Specific comments about the manuscript
line 81: "...into two levels: micro and macro." It sounds for me that noun is missing from here e.g. "micro and macro scales"
line 86: if you used the scientific name for the South China Tiger, then use the sci. name for the bear species too or use the (Ursidae) term here from line 96
line 122: One very important information is missing: the estimated population density of wild boar. At least some numbers about the population size, and distribution is needed, but county-scale density data would be the best, visualized on a map. On the second hand, readers don’t know anything whether wild boar is actively controlled in China by hunting or other methods. Is there any active monitoring system that monitors the population trend? What is the spatial distribution of the species in the country? Is there any significant spatial difference among specific regions in boar density?
line 133: Authors should handle damages to property and incidents with humans separately. I understand that these are both belong into the human-wildlife conflicts, but the purpose of this study is to reveal where wild boar can cause damages to property and which regions are the potential danger zones where direct attacks has higher probability! If it is possible I recommend to report some thoughts about that in the manuscript at least.
line 134: “…used the Houyi collector…” Citation/reference needed
line 138: Citation/reference needed for Baidu Maps
line 139: Citation/reference needed for ArcGIS, which version number and which toolbox was used?
line 140-141: “…locations, ensuring that each raster had only one distribution location and that the distance between every two locations was more than 10 km.” It sounds a bit unclear. That means the biggest resolution (raster size) was 10km? (>=10km)
line 144: Citation/reference needed for Maxent model if available. Which software was used to perform modelling?
line 190: “…since Phillips wrote and developed the model using JAVA language in 2004…” Citation/reference needed
line 209: Citation/reference needed for the used Maxent model, and the same for the used software.
line 212: Were there any highly positive outliers? (E.g. places where many incidents/damages were reported) this would be an interesting additional result.
line 239-243: These lines belong to the discussion chapter.
line 253: the term "the maximum values" is not so clear here, 16C and 800 mm refers to the highest probability when wild boar related incidents can occur right? Clarify this sentence.
line 254: I find it unnecessary to explain complicated curve shapes with a letter in all cases. The high increase just occurred due to low (or none) data from areas near to sea level or under sea level. This tells us nothing about the probability of conflicts, rather call it "bimodal" or use a similar term.
line 255-256: This belongs to the discussion, furthermore it needs to be explained in detail, since it does not mean that conflicts cannot emerge between these peak points.
line 258-262: This belongs to the discussion. On the other hand, many studies state that boars (and other ungulates) tend to forage in croplands because they can find shelter, cover and easily accessible food there. I think that the insufficient natural food source is a bit unfounded in this case; or cite more proof.
line 264-269: This belongs to the discussion, and use relevant citations!
line 284-292: This belongs to the discussion.
line 288-291: I agree that higher population density and development increase the risk, but cannot believe that there was no boar incidents on the western side of the country. Since data points were missing from there, and no direct surveys were done, Authors need to be more cautious, when stating these and similar thoughts.
line 302-306: This belongs to the discussion.
line 305-306: „devastated by” this term sounds strange here, It suggests that the complete elimination of wild boars is the only solution to the problem, and there is no proper management. Please clarify this sentence.
line 311-314: Redundancy, it was mentioned in the methods already.
line 327-328: In China or in general? If latter then provide citations.
line 334-338: Too long and too general, already reported in the methods.
line 339-341: This belongs to the data and methods chapter.
line 355-357: Place this sentence before the sentence in line 353-355 and keep the starting words (Our results showed...)
line 361: “…indirectly affecting the range and frequency of wild boars' activities.” Provide relevant citations!
line 364-366: This is not so clear, e.g. livestock is highly related to cultivated land, and this can provoke depredation by carnivores! Furthermore, in the discussion section Autors need to verify and contrast their results with other relevant studies. This is missing from here, as well as the citations!
line 375-376: Please provide more insight about the current situation, and create plans/ideas how these type of studies can support the decision making systems, what can be improved in the study, and what other factors can be enhance a model like this to create a successful forecasting system? I think some practical recommendations to stakeholders and decision makers would greatly improve this chapter (e.g. hunting/culling based on these predictions in the future, or contribute to monitor boar population density). This can improve the message of the study.
line 381: “…and cannot be generalized.” Well black box models like this are for generalizing. Considering this, Authors should emphasize, why machine learning models like this are an adequate (but not satisfactory) tools for this problem! - e.g. using weights in data evaluation should improve prediction and decrease overfitting (eg. consider to add wild boar population densities to the model)
The text was written in high quality English, however some words should be replaced (see my comments and suggestions).
Reviewer 2 Report
Boming Zheng et al. analyzed and identified risk regions of wild boar (Sus scrofa) incident in China. They found that approximately 12.18% of the area in China is at high risk of wild boar incidents occurring, mainly on the eastern side of the Huhuanyong Line. They do it on a large scale by collecting existing data from 2008-2022. Although the significance of this article is low, this finding is helpful in identifying risk regions of wild boar to some extent.
In addition, the wild boar incident locations were mainly collected from Middle East in china (fig.1), and they concluded that the high risk of wild boar incidents occurring mainly on the eastern side of the Huhuanyong Line. I think this conclusion is predictable because of geography and the way the authors collected samples. Overall, I advise authors to submit to other journals.
Reviewer 3 Report
Please see the attachment.

Round 2
Reviewer 2 Report
The author has made a lot of changes and seems to have addressed my concerns
Reviewer 3 Report
The manuscript is sufficiently improved.